# Free Radical Isomerizations in Acetylene Bromoboration Reaction

**DOI:** 10.3390/molecules26092501

**Published:** 2021-04-25

**Authors:** Hugo Semrád, Ctibor Mazal, Markéta Munzarová

**Affiliations:** Department of Chemistry, Faculty of Science, Masaryk University, Kotlářská 2, 611 37 Brno, Czech Republic; semrad.hugo@mail.muni.cz (H.S.); mazal@chemi.muni.cz (C.M.)

**Keywords:** free radicals, isomerization, acetylene, bromoboration, mechanism, addition-elimination, DFT, MP2

## Abstract

The experimentally motivated question of the acetylene bromoboration mechanism was addressed in order to suggest possible radical isomerization pathways for the *syn*-adduct. Addition–elimination mechanisms starting with a bromine radical attack at the “bromine end” or the “boron end” of the C=C bond were considered. Dispersion-corrected DFT and MP2 methods with the SMD solvation model were employed using three all-electron bases as well as the ECP28MWB ansatz. The rate-determining, elimination step had a higher activation energy (12 kcal mol^−1^) in case of the “bromine end” attack due to intermediate stabilization at both the MP2 and DFT levels. In case of the “boron end” attack, two modes of C–C bond rotation were followed and striking differences in MP2 vs. DFT potential energy surfaces were observed. Employing MP2, addition was followed by either a 180° rotation through an eclipsed conformation of vicinal bromine atoms or by an opposite rotation avoiding that conformation, with 5 kcal mol^−1^ of elimination activation energy. Within B3LYP, the addition and rotation proceeded simultaneously, with a 9 (7) kcal mol^−1^ barrier for rotation involving (avoiding) eclipsed conformation of vicinal bromines. For weakly bound complexes, ZPE corrections with MP2 revealed significant artifacts when diffuse bases were included, which must be considered in the Gibbs free energy profile interpretation.

## 1. Introduction

Alkyne bromoboration is an electrophilic addition of BBr_3_ or other bromoboranes to an alkyne triple bond. When boron tribromide is used, binding the BBr_2_ group and Br atom to opposite ends of the C≡C bond results in either a (*Z*)- or (*E*)-adduct. In the first experimental report from Lappert and Prokai from 1964 [1], a theoretically interesting difference was noticed in the stereochemistry of products for the case of addition to diphenylacetylene vs. acetylene. While bromoboration of the former provided the *syn*-adduct, the reaction of the latter resulted in the thermodynamically more stable *anti*-adduct. Two hypotheses were introduced to interpret the result: there is a different mechanism for addition to acetylene, missing a four-centered transition state present for substituted acetylenes, or there is a thermodynamic rather than kinetic control of haloboration reactions. A follow-up work of Blackborrow from 1973 suggested that *Z*- and *E*- additions occur by different mechanisms under kinetic control, with *E*-products favored by solvents more readily supporting polar transition states [2]. The kinetics of bromoboration were then studied in detail for a series of acetylene derivatives with two ethyl, propyl, an one or two butyl substituents, with a 15:1 *Z/E* adducts ratio found after 1 h from preparation and a 1:7 *Z/E* ratio found after a few days [3]. In 1994, by a reaction of BBr_3_ with acetylene and subsequent esterification with diisopropyl ether, Mazal and Vaultier developed a straightforward stereoselective synthetic route for 1-(dialkoxyboryl)-1,3-dienes [4].

The bromoboration reaction fully manifested its application potential in the stereoselective syntheses of alkenes developed by Suzuki in the late 1980s [5]. A complete account of haloboration work up to 2020 was reviewed by Kirschner, Yuan, and Ingleson [6], which also summarized the three most recent contributions to haloboration of simple alkynes. A 2012 computational study by Wang and Uchyiama concentrated on mechanistic aspects of the reaction [7]. A 2013 joint experimental and theoretical study by Lawson et al. reported the first successful haloboration/esterification of internal alkynes [8]. Finally, inspired by the previous work by Mazal and Vaultier [4], we concentrated on the stereoselective bromoboration of acetylene with boron tribromide as a route to (*Z*)-Bromovinylboronates, with mechanistic studies supported by ab initio calculations [9].

In the latter study, we suggested—besides a *syn*-addition through a four-centered transition state—two other addition mechanisms yielding a thermodynamically more stable (*E*)-isomer [9]. We further concentrated on the mechanism of *Z*/*E*-isomerization of the *syn*-adduct, (*Z*)-dibromo(2-bromvinyl)borane, (*Z*)-**1**, considering Wang and Uchyiama’s hypothesis of stereoconversion mediated by BBr_3_ or HBr. BBr_3_ was taken for a catalyst of isomerization in many previous works [1,10,11]. Since BBr_3_-catalyzed isomerization with relatively high activation Gibbs energy (30 kcal mol^−1^, [7]) did not apply under our experimental setup, but the catalytic effect of HBr was confirmed [9], we concentrated on understanding its mechanistic aspects. Knowing that, for the chloro-counterpart of (*Z*)-**1**, a direct *cis*/*trans* isomerization and HCl-catalyzed conversion required as much as 55 and 45 kcal mol^−1^, respectively [7], we analyzed several HBr-catalyzed mechanisms for (*Z*)-**1** with a hope to establish a less-energy demanding route. Our results summarized in Appendix A suggest that a polar HBr-catalyzed addition–elimination stereoconversion is highly unlikely to proceed.

Thus, we concentrated on radical *Z/E* isomerization possibility. As formation of the vicinal isomer of dibromoethane in the reaction mixture indicated that HBr prefers a radical pathway for the addition to vinyl bromide in BBr_3_, we suggested a free radical mechanism of *Z/E* stereoconversion [9]. Several methodological questions arose, however, which were beyond the scope of our previous study; thus we formulate and respond them here: (1) Could our earlier MP2 radical isomerization reaction profiles be obtained at the DFT level of theory as well? (2) How strongly are electronic energies, zero-point energies, thermochemical corrections, and entropy contributions dependent on the basis set employed and/or on the relativistic pseudopotential bromine core electrons treatment? (3) What is the exact reason for some intermediates being located in Gibbs free energy above related transition states; to what extent is this counterintuitive behavior dependent on the methodology, and to what extent is this a footprint of PES flatness for radical reactions?

To respond these questions, we undertook a series of MP2 and B3LYP calculations, modeling two possible attacks of a bromine radical on the C=C double bond of (*Z*)-**1**. We then searched for mechanisms leading to such product complexes of (*E*)-**1** with Br radical, which are located in Gibbs free energy below the starting reactant complex of (*Z*)-**1** and Br radical. Figure 1, relating the frontier orbitals of (*Z*)-**1** to fragment molecular orbitals of BBr_3_ and acetylene, illustrates which sites at (*Z*)-**1** were considered for a bromine radical attack: First, Br^•^ can act as an electrophile and interact with the HOMO of alkene at the site of carbon bound to bromine (“bromine end”) of (*Z*)-**1**. Second, Br^•^ can act as a nucleophile and interact with the LUMO of alkene at the site where it carbon bound to boron (“boron end”). The following step is a rotation about the C–C bond, which can proceed in two directions that we classify below, as described in Scheme 1b. The following step is a rotation about the C–C bond which can proceed in two directions. The rotation involving an eclipsed conformation of vicinal bromine atoms is termed as “clockwise” according to orientation of the structures used in all presented schemes, while the rotation in the opposite direction is termed as “anticlockwise” (see Scheme 1b). The paper is organized as follows: We start with reviewing the first addition step of the bromoboration reaction studied earlier in Reference [7], and augment it with basis set dependence and B3LYP results. Then, we consider the “bromine end” attack and the “boron end” attack with method order (MP2 or DFT first), reflecting the genuine progress of our potential energy surface explorations.

## 2. Results and Discussion

### 2.1. Addition Step of the Bromoboration Reaction

The addition of BBr_3_ on C_2_H_2_ is most easily described as a Lewis acid (BBr_3_) attack on a Lewis base (acetylene triple bond) or vice versa. From a qualitative theory of molecular orbitals point of view, it is an interaction between a doubly degenerate HOMO of the alkyne with the LUMO of boron tribromide, cf. Scheme 1. The reaction itself starts from a weakly bound reactant complex of C_2_H_2_ and BBr_3_, stabilized by a π-hole–π-electrons interaction, referred to as the triel bond [12], and related in its nature to a hydrogen bond [13]. The reaction profile for the following single-step addition leading to the *syn*-adduct, **(*Z*)-1,** is shown in Figure 1. The reaction proceeded via a four-centered transition state (**TS1** in Figure 1), as reported by Wang and Uchiyama [7]. Compared to MP2 results obtained here and in Reference [7], B3LYP predicted lower activation barriers, by 4 kcal mol^−1^ with the aug-cc-pVTZ basis and by 10 kcal mol^−1^ with the 6-31+G*/SVP basis. At the same time, B3LYP estimated higher thermodynamic stabilization of the *syn*-adduct with respect to reactants: by 3 kcal mol^−1^ (aug-cc-pVTZ) to 8 kcal mol^−1^ (6-31+G*/SVP) basis. 

### 2.2. Isomerization through “Bromine End” Attack

#### 2.2.1. B3LYP Results

We started the isomerization studies with the more economical B3LYP approach, using the 6-31+G*/SVP basis, two basis sets of TZP quality, and the quasi-relativistic ECP treatment of bromines combined with 6-31+G* for light atoms. Total electronic energy and nuclear repulsion profile (Figure 2) started from **(*Z*)-1 + Br^•^** with a saturation of bromine-bearing carbon (C_Br_) to give **IM1**. It then continued with a rotation of the CHBr_2_ group (cf. Scheme 1b) to **IM2** and, through a release of a **Br^•^** radical, proceeded to a first product complex, **(*E*)-1 + Br^•^**. The weak coordination of **Br^•^** in **(*Z*)-1 + Br^•^** and **(*E*)-1 + Br^•^** is described in Scheme 2. Activation barriers were, apart from the last step, dominated by the C–C rotation, requiring only 5 kcal mol^−1^ with all basis sets. 

Figure 3 displays Gibbs free energy profile, which, apart from total electronic energy, also includes the zero-point energy (ZPE), thermal corrections to enthalpy, and entropy contributions at 298 K. These were obtained with the same basis sets as the respective electronic energies. In terms of total electronic energy (denoted below as Δ*E*) and also Gibbs free energy (denoted below as Δ*G*), **(*E*)-1 + Br^•^** was more stable than **(*Z*)-1 + Br^•^**. Thus, the three-step mechanism just described could end here, describing a thermodynamically spontaneous process. However, in case of MP2 this was not so. This is why, for a comparison of B3LYP and MP2 methodology, an additional step was considered, where a new local minimum denoted as **^add^(*E*)-1 + Br^•^** was obtained. The latter was established by scanning Δ*E* with respect to C_Br_ ‒ Br^•^ bond prolongation, establishing the point of maximum energy as a TS estimate, and proceeding via optimization into TS and IRC procedures to both corresponding minima. In **^add^(*E*)-1 + Br^•^**, **Br^•^** coordination to C_Br_ was much weaker (cf. Scheme 2) and the radical could be already considered to be a free one. The activation barrier of 8–9 kcal mol^−1^ dominated the whole mechanism, but should be significantly decreased if a **Br^•^** radical would be directly transferred to a “fresh” **(*Z*)-1** molecule to enter a new isomerization cycle. 

In terms of Δ*G* (Figure 3), the reaction profile predicted **(*E*)-1 + Br^•^** to be again the most stable species of the mechanism, stabilized with respect to **(*Z*)-1 + Br^•^** by ca. 3 kcal mol^−1^ for all basis sets. This was to be expected due to the presumably stronger interaction energy in the complex with **Br^•^** lying closer to the π-electron system than in the case of **^add^(*E*)-1 + Br^•^**. Quantitatively, the Δ*E* and Δ*G* profiles differed in the relative barriers of addition, rotation, and elimination steps, as well as in the basis set influence. The Δ*E* profile (Figure 3) was less basis set-dependent, with the addition step dominating the energy requirements. The Gibbs energy profile was more basis-set sensitive than the Δ*E* profile, with the flattest PES predicted by ECP treatment, and the rate-determining step again being the rotation one.

#### 2.2.2. MP2 Results

The reaction profiles for Δ*E* and Δ*G* with MP2 are shown in Figure 4 and Figure 5, respectively. Please note that, unlike in the case of the B3LYP results, all MP2 contributions to Δ*G* beyond Δ*E* (in Figure 5) were covered using the smallest, 6-31+G*/SVP basis set. Reasons for doing so are explained below, while the energy profiles including ZPE and other corrections using the same basis for obtaining electronic energies are shown in Appendix A. MP2 reaction profiles followed the same reaction course as with B3LYP, but the activation barriers were substantially different. MP2 profiles for Δ*E* and Δ*G* were dominated by **Br^•^** elimination from **IM2** (12 kcal mol^−1^ in both cases), which was more than doubled with respect to the B3LYP results (5 kcal mol^−1^ in Δ*E* and Δ*G* for all-electron bases, 2 kcal mol^−1^ in case of ECPs). On the contrary, the rotation barrier was almost identical for MP2 and B3LYP (4–5 kcal mol^−1^ in Δ*E*, 6 kcal mol^−1^ in Δ*G*). 

The most striking difference between Figure 4 and Figure 5 is the unexpectedly high destabilization of **(*E*)-1 + Br^•^** with respect to all other PES local energy minima. With the 6-31+G*/SVP basis, **(*E*)-1 + Br^•^** had higher Gibbs energy than **TS^add^**. This counterintuitive result was even more pronounced in case of the diffuse bases if these were employed for ZPE and other corrections to Δ*G*, cf. Appendix A. After carefully checking a tight convergence of both the IRC and optimization procedures, and excluding a possible reason in the basis set superposition error, we identified unexpectedly large ZPE values as the origin of the **(*E*)-1 + Br^•^** instability. Examples of this seldom behavior can be found in the literature [14]. 

Table 1 summarizes the zero-point energy values for the mechanisms shown in Figure 2, Figure 3, Figure 4 and Figure 5. With B3LYP, ZPEs lay in a narrow region of 23.9–24.8 kcal mol^−1^ for all stationary points and basis sets. With MP2 and 6-31+G*/SVP basis, ZPEs ranged from 24.6 to 25.3 kcal mol^−1^ for the majority of local minima and saddle points, but had clear outliers in **(*Z*)-1 + Br^•^**, **(*E*)-1 + Br^•^**, and **TS^add^**. While for **(*E*)-1 + Br^•^** and **TS^add^**, ZPE values decreased with larger basis sets, for **(*Z*)-1 + Br^•^** with aug-cc-pVTZ, and especially for Def2TZVPP, these grew further, up to 40 and 53 kcal mol^−1^, respectively, which was obviously a computational artifact. As suggested by a reviewer, we checked whether, for the weakly bound complexes, an application of frequency scale factors [15] might reduce ZPVE contributions. However, the influence of scaling on the latter was close to negligible from the point of view of reaction energy profile, cf. Table 1.

Since a detailed analysis and a possible cure for the unfortunate performance of MP2 for ZPE estimation was beyond the scope of the current study, we represented MP2 energy profiles using the most acceptable ZPE values obtained with the 6-31+G*/SVP basis. However, we expect that these ZPEs were still overestimated, since in Δ*G*, **(*E*)-1 + Br^•^** lies 1.5 kcal mol^−1^ above **(*Z*)-1 + Br^•^,** while **(*E*)-1** lies below **(*Z*)-1**, see [7]. A still more intuitively acceptable, but less theoretically consistent profile, would be obtained when using DFT values for ZPE corrections of top of MP2 electronic energy profiles. Instead, we supplemented the mechanism by an additional step in an effort to find a more stable local minimum denoted as **^add^(*E*)-1 + Br^•^**, where “add” stands for “additional (step)”. For the total electronic energy profile (and at the B3LYP level for both profiles), the latter step was redundant for the thermodynamically spontaneous reaction course description.

A successful application of frequency scale factors presumes a relatively uniform nature of the overestimation of quantum chemical vibrational frequencies [15]. We compared individual vibrations for **(*E*)-1 + Br^•^** and **^add^(*E*)-1 + Br^•^** with B3LYP and MP2 methods in order to see whether MP2 frequencies for **(*E*)-1 + Br^•^** were overestimated consistently or revealed significant outliers. The results in Appendix A prove the latter option. While for **^add^(*E*)-1 + Br^•^**/B3LYP, **^add^(*E*)-1 + Br^•^**/MP2, and **^add^(*E*)-1 + Br^•^**/MP2, the two highest frequencies lay at 3190–3270 cm^−1^ and corresponded to individual in-plane vibrations of the two C-H bonds, in the case of **(*E*)-1 + Br^•^**/MP2, these were overreached by a 6900 cm^−1^ vibration coupling out-of-plane hydrogen and carbon movements. We believe that this was related to spin density distribution in **^add^(*E*)-1 + Br^•^**/MP2, which is compared to spin density distributions of other structures discussed above in Scheme 3. As seen already from the structural parameters in Scheme 2 and Scheme 4, the coordination of **Br^•^** to alkene was stronger for **(*E*)-1 + Br^•^** than for **^add^(*E*)-1 + Br^•^** with both B3LYP and MP2. This was reflected already in the C=C bond lengths. However, while in the case of B3LYP, spin density was shared between **Br^•^** and the more distant carbon atom, in the case of MP2, the spin density delocalized to both carbons equally. We expect that this caused a complex mixing of vibrational modes and their extreme sensitivity to the method and basis set employed. 

### 2.3. Isomerization through the “Boron End” Attack with Clockwise Rotation

#### 2.3.1. MP2 Results

The second pathway is represented by a bromine radical addition to the carbon carrying the dibromoborane group. In contrast to the “bromine end” mechanism, activation barriers depend on the modes of rotation, since these start from a non-planar intermediate. We opened this discussion with the clockwise rotation route and MP2 results, since the stationary points of PES were easier to identify than with B3LYP, see Section 2.3.2. The reaction profile in Δ*E* (Figure 6) started with the formation of **IM3**, which was then clockwise-rotated into **IM4**, and released **Br^•^** to form **(*E*)-1 + Br^•^**. The additional step loosened the weak coordination of **Br^•^** and was included for same reasons as discussed in Section 2.2. 

The Gibbs free energy profile (Figure 7) was similar to the “bromine end” attack one (Figure 5) in the sense of problematic ZPE corrections, which are listed in Table 2. Again, all contributions to Δ*G* beyond Δ*E* were covered using the 6-31+G*/SVP basis, irrespective of the basis set used for Δ*E* determination. As a result, **(*Z*)-1 + Br^•^** was significantly stabilized with respect to the larger basis set ZPE results, and **TS2′** was stabilized even more. Consequently, the **(*Z*)-1 + Br^•^** reactant complex was justly found lower in Δ*G* than were the isolated reactants, and the following activation barrier was reduced. The product complex was again +1.5 kcal mol^−1^ higher in Δ*G* than the reactant complex, and it was even 3–5 kcal mol^−1^ higher (depending on the basis set for Δ*E*) than **TS4′**. Interestingly, the elimination barrier for the cleavage of the bromine radical from **IM4** was 2–3 times lower than in the case of the “bromine end” route. The last step of the mechanism again illustrated the effort to find a more stable local minimum than **(*E*)-1 + Br^•^** at the level of theory employed.

#### 2.3.2. B3LYP Results

While in the case of the bromine end attack, the transition states on the B3LYP reaction profile were easily localized using the single-coordinate-driving method (cf. Section 3), it was not so in the case of the boron end attack. In particular, a B3LYP estimate of **TS2′** from Figure 6 and Figure 7**,** either obtained from the B3LYP scans of PES or the B3LYP optimized from MP2 geometry, always decomposed into **(*Z*)-1 + Br^•^** in either direction of the IRC procedure. On the contrary, **TS3′** determined by MP2 was easily reoptimized into its B3LYP counterpart and, by means of IRC, proved to connect **(*Z*)-1 + Br^•^** with **(*E*)-1 + Br^•^**. As shown in Figure 8, this happened in a single addition-rotation step with a 9 kcal mol^−1^ activation barrier in Δ*G*, followed by a barrierless elimination. Even though no intermediate was present between **(*Z*)-1 + Br^•^ and TS3′**, the IRC coordinate possessed two distinct phases: (a) an addition connected with only slight rotation and (b) the rotation phase itself. Such phase-like reaction profiles have been reported earlier for mechanisms of barrierless reactions [17]. Thus, the pattern of very flat potential energy surfaces observed sometimes for closed-shell [18], and much more often for radical reactions [19,20,21], in our case was stressed by B3LYP, where it led to a disappearance of particular intermediates from the reaction coordinate. 

### 2.4. Isomerization through the “Boron End” Attack with Anticlockwise Rotation

#### 2.4.1. MP2 Results

The last proposed mechanism converting **(*Z*)-1 + Br^•^** into **(*E*)-1 + Br^•^** differed from mechanism described in Section 2.3.1 by an opposite direction of rotation around the C–C bond. The total electronic energy profile and Gibbs free energy profile are shown in Figure 9 and Figure 10, respectively. Again, ZPE corrections for the addition and elimination step were significantly overestimated with the diffuse basis sets, cf. Table 2. Hence, Figure 10 reports all contributions beyond the total electronic energy with the smallest, 6-31+G*/SVP basis set, while ZPE, thermal corrections to enthalpy, and entropy terms with the corresponding diffuse basis sets are shown and accounted for in Appendix A. Note that the conformational transition from the local minimum **IM3** to **IM4** in Figure 10 was associated with a negligible activation barrier.

#### 2.4.2. B3LYP Results

Like in Section 2.3.2, also in the case of the boron-end attack plus anticlockwise rotation, the MP2 reaction profile was successfully accomplished prior to the DFT one. It was then employed for identifying stationary points of B3LYP potential energy surface, which again connected the addition, rotation, and eliminations steps into what was effectively a single-step transformation. Indeed, the barriers between **IM3a**, **TS3′a**, and **TS3′b** were negligibly small. The B3LYP approach thus again simplified the proposed mechanisms, and did so with a smaller activation barrier for this anticlockwise rotation (7 kcal mol^−1^) than reported in the case of the clockwise rotation (9 kcal mol^−1^, cf. Section 2.3.2).

## 3. Computational Details

### 3.1. Starting Structures, PES Stationary Point Determination, and Their Verification

Starting structures representing reactants and/or products were prepared and preoptimized using the UFF method within the Avogadro 1.1.1 software [22,23]. The resulting geometries were optimized using ab initio methods and the basis sets specified below. The single-coordinate-driving method was employed for scanning the potential energy surfaces in order to obtain transition state structure estimates [24,25], which were then optimized. All structures of local minima and transition states were verified by the harmonic vibrational frequency calculations. The connectedness of transition states with corresponding local minima was confirmed by both inspection of the single imaginary frequency motion and IRC calculations in both directions [26]. 

### 3.2. Ab Initio Methods, Basis Sets, Pseudopotentials, and Solvent Model

All structures and their total energy contributions were obtained at either the B3LYP level [27,28] with Grimme’s GD3BJ dispersion correction [29,30] or at the MP2 level [31]. Three all-electron basis sets were employed for the B3LYP and MP2 calculations: Pople-style 6-31+G* basis for light atoms [32,33,34] combined with Ahlrich’s SVP all-electron basis for bromine [35], Ahlrich’s Def2TZVPP basis for all atoms [36], and Dunning’s aug-cc-pVTZ basis for all atoms [37]. While the first basis was selected for the sake of comparison with Reference [7], the diffuse bases were tested in order to enable proper treatment of dispersion interactions at the MP2 level of theory [38]. In the case of B3LYP, comparative calculations were done employing the ECP28MWB quasi-relativistic effective core potentials for bromine atoms combined with the corresponding DZ orbital basis [39] and with the all-electron 6-31+G* bases for the light atoms. Implicit solvation in CH_2_Cl_2_ was included using the SMD continuum solvent model [40]. 

### 3.3. Implementation, Energy Contribution Representations, and Structure Visualizations

All calculations were performed in the implementation of Gaussian 09, rev. D.01 [41]. An ultrafine integration grid as well as tight optimization criteria were employed for both self-consistent-field and optimization procedures. The sums of electronic and thermal free energies as well as individual contributions into these are summarized in Appendix A. Total electronic energies and Gibbs free energies were plotted along reaction coordinates, shown in Figure 1, Figure 2, Figure 3, Figure 4, Figure 5, Figure 6, Figure 7, Figure 8, Figure 9, Figure 10 and Figure 11. Visualizations of structures in Figure 1, Figure 2, Figure 3, Figure 4, Figure 5, Figure 6, Figure 7, Figure 8, Figure 9, Figure 10 and Figure 11 and Scheme 1 and Scheme 3 were done using Molden 5.2 [42]. Scheme 2 and Scheme 4 were prepared using Chemdraw 20.0.

## 4. Conclusions

The present study shows that, out of two possible sites for radical attack on the C=C bond, the one on the “bromine end” corresponded to a *Z*/*E* isomerization mechanism with a larger elimination barrier due to more pronounced intermediate stabilization. B3LYP and MP2 approaches provided the same qualitative characteristics of potential energy surface in this case. On the contrary, in the case of the “boron end” radical attack, intermediates were less stable and the B3LYP vs. MP2 reaction profiles were qualitatively different. While MP2 predicted that the addition, rotation, and elimination steps of the isomerization would proceed consecutively, B3LYP found these processes to be concerted. Unlike the “bromine end” mechanism, an additional degree of freedom entered the “boron end” route in terms of direction of the rotation step. The two rotation directions thus differ in the number of steps necessary for *Z*/*E* isomerization, but are comparable in terms of energy requirements.

It is beyond the scope of the current study to predict whether the concerted, B3LYP mechanism or the consecutive, MP2 mechanism represent the true reaction course in the case of the “boron end” radical attack. The ultimate decision would have to be based on experimental data on the presence or absence of radical intermediates, which could be provided by electron paramagnetic resonance measurements. An additional option is a combination of B3LYP and MP2 with other quantum chemical methodology, such as the CASSCF and CASPT2 methods, used with success for radical reaction energetics calculations by Zipse [43]. 

Definitely the most serious computational drawback demonstrated by this study was that of the ZPE overestimation for systems with a weakly coordinated bromine radical by the MP2 method. This happened especially for the basis sets that were rich in diffuse functions; however, even the smallest, 6-31+G*/SVP basis set provided too high ZPE corrections for **(*E*)-1 + Br^•^** to give the correct thermodynamical ordering of *Z*/*E* isomers. On the contrary, the B3LYP approach was much more robust in terms of the ZPE results. While we circumvented the problem of artificially high ZPE corrections by finding another PES local minimum for **(*E*)-1 + Br^•^** with “a correct” thermodynamical position with respect to **(*Z*)-1 + Br^•^**, future work is required in order to understand and solve this methodological problem.

Despite the theoretical drawbacks described, the radical mechanisms proposed in this work are consistent with available experimental data [9]. They thus offer a novel point of view on the mechanism of bromoboration and haloboration reactions and, in a wider context, on the isomerizations of substituted alkenes in general. 

## Data Availability

Data supporting reported results are available online.

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
