# Peer review of "Free Radical Isomerizations in Acetylene Bromoboration Reaction"

_molecules, 2021, doi:10.3390/molecules26092501_

Round 1
Reviewer 1 Report
The authors described a theoretical study on the reaction mechanism of free radical isomerizations in acetylene bromoboration. Different computational methods were employed and compared. The results were well presented and convincing. The computational approach does have limitations, which need future experimental investigation to reveal more details of the reaction mechanism. However, the manuscript provides a solid basis for future research. It is recommended to publish in the current form.
Author Response
We are pleased by the positive response of Reviewer 1. There weren't any comments to be addressed.
Reviewer 2 Report
This is a theoretical study of the bromoboration of acetylene, followed by
Z/E-isomerization of the adduct, with two computational methods (B3LYP/MP2),
which was previously investigated by MP2 only (Ref. 9). The mechanism of
the isomerization catalyzed by a Br radical was investigated by locating
transition states and intermediates of the reaction.
The outcome of this study is neither surprising nor conclusive, and does
not add a lot of new insight to the results presented in ref. 9.
Basis set effects are more or less negligible. It is trivial that
larger basis sets will improve results, and it has been established
many times that small basis sets may improve results of B3LYP by
error compensation.
One could try to correct the observed overestimation of ZPE with MP2
(line 196) by scaling the vibrational frequencies or the ZPE correction.
There should be a certain error due to the harmonic approximation
which breaks down esp. for the low frequency vibrations in the
weakly coordinating complexes. Please check the according literature
(e.g. Radom J. Phys. Chem. A 2007, 11683) whether applying a scaling
factor could improve the results.
The DFT results for the Br radical attack at the "boron end" (wrongly
assigned in lines 85-87) contradict the MP2 PES and predict a single
step addition/rotation/dissociation while several intermediates are
found with MP2 along the the C-C bond rotation. The comparison of
the two methods is not really conclusive as a high-quality method
like DLPNO-CCSD(T) is missing here which could indicate what is closer
to the physical reality. DLPNO-CCSDT) calculations are not impossible
for these small radicals.
Finally, what is the electronic nature of "(E)-1+Br." and "add(E)-1+Br." ?
The conversion could be characterized as 'bond length' isomerism, but
obviously the nature of bonding is different. Is the single electron
more delocalized in "(E)-1+Br." than in the "add(E)-1+Br."?
Please plot the spin density of the two complexes, this could help
to elucidate the nature of the two complexes.
(There should also a "add(Z)-1 + Br." complex, I suppose.)
Author Response
- The outcome of this study is neither surprising nor conclusive ... and it has been established many times that small basis sets may improve results of B3LYP by error compensation. We agree with the reviewer that many outcomes are far from being surprising. Yet we believe that some methodological aspects (especially those regarding DFT vs. MP2 reaction profiles) which could not be presented and discussed in ref. 9 raise interesting points which can support further development in the field.
- One could try to correct ... by scaling the vibrational frequencies or the ZPE correction.
We thank the reviewer for suggesting this. We checked the suggested literature and augmented ZPE values in Tables 1 and 2 by their scaled values. The improvement of the results was unfortunately negligible, thus we analyzed the individual frequencies and noted an outlier for the (E)-1+Br. at MP2 level. Corresponding data are shown in newly inserted tables S10 and S11 and analyzed on lines 207-250 of the revised manuscript. - The DFT results for the Br radical attack at the "boron end" (wrongly assigned in lines 85-87) contradict the MP2 PES ....The comparison of
the two methods is not really conclusive as a high-quality method
like DLPNO-CCSD(T) is missing here... We have corrected the assignment in lines 85-87. We agree that the comparison of the two methods is inconclusive. However, even though submitted sample DLPNO-CCSD(T) calculations immediately after receiving the report, any conclusive results could not be obtained within the time limit for the revision. We hope we can understand this constructive point made by the reviewer as advice for our future work. - Is the single electron more delocalized in "(E)-1+Br." than in the "add(E)-1+Br."? Please plot the spin density of the two complexes, this could help to elucidate the nature of the two complexes.We augmented the manuscript with new Schemes 3 and 4, where the latter plots the spin densities and the former summarizes key structural parameters of "(E)-1+Br." and "add(E)-1+Br." at MP2 level. The corresponding discussion can be found in lines 237-246. The spin density is more delocalized for "(E)-1+Br.", but differently with B3LYP than with MP2. We speculate that this may be the reason why "(E)-1+Br." frequencies are difficult to calculate with MP2 but not so much with B3LYP.
Round 2
Reviewer 2 Report
The authors have taken up the points which I had raised in my first report. From my viewpoint, the paper now has grown in quality and should be published in the present form.